# The Long Telling Story of “Endothelial Progenitor Cells”: Where Are We at Now?

**DOI:** 10.3390/cells12010112

**Published:** 2022-12-28

**Authors:** Maria Cristina Vinci, Ermes Carulli, Erica Rurali, Raffaella Rinaldi, Giulia Damiano, Angela Raucci, Giulio Pompilio, Stefano Genovese

**Affiliations:** 1Unit of Vascular Biology and Regenerative Medicine, Centro Cardiologico Monzino IRCCS, 20138 Milano, Italy; 2Scuola di Dottorato in Medicina Traslazionale, Università di Milano, 20100 Milano, Italy; 3Dipartimento di Scienze Cliniche e di Comunità, Università di Milano, 20100 Milano, Italy; 4Unit of Experimental Cardio-Oncology and Cardiovascular Aging, Centro Cardiologico Monzino IRCCS, 20138 Milano, Italy; 5Dipartimento di Scienze Biomediche, Chirurgiche e Odontoiatriche, Università degli Studi di Milano, 20100 Milano, Italy; 6Diabetes, Endocrine and Metabolic Diseases Unit, Centro Cardiologico Monzino IRCCS, 20138 Milano, Italy

**Keywords:** endothelial progenitor cells, CD34^+^ hematopoietic stem progenitor cells, regenerative medicine, cardiovascular disease, angiogenesis

## Abstract

Endothelial progenitor cells (EPCs): The name embodies years of research and clinical expectations, but where are we now? Do these cells really represent the El Dorado of regenerative medicine? Here, past and recent literature about this eclectic, still unknown and therefore fascinating cell population will be discussed. This review will take the reader through a temporal journey that, from the first discovery, will pass through years of research devoted to attempts at their definition and understanding their biology in health and disease, ending with the most recent evidence about their pathobiological role in cardiovascular disease and their recent applications in regenerative medicine.

## 1. Introduction

In the late 1990s the discovery of endothelial progenitor cells (EPCs) has revolutionized paradigms of physiology and pathology. For the first time, the existence of this cell population in the peripheral blood demonstrated that both vasculogenesis and angiogenesis could arise simultaneously during adulthood, overturning the dogma that vasculogenesis only could occur during embryogenesis [1]. The discovery opened a new era as it posed the basis for a theoretical regeneration of the cardiovascular (CV) system. As expected, on this wave of the novelty, these cells were subject to intense basic, preclinical, and clinical research that led to the accumulation of numerous publications. However, despite the relevant number of studies, the findings with regard to EPC origin and biological characteristics were controversial, ambiguous, and raised confusion in the field. The lack of consensus mainly arose from the heterogeneous population of cells comprised in the term EPCs, and on the different antigens, protocols, and cell culture methods used for their identification isolation and characterization.

Although it is now clear that there is no specific marker able to uniquely identify EPCs, this has not undermined the proof of their existence. Beyond the initial enthusiasm, the EPCs and especially the cells from which they originate, the CD34^+^ hematopoietic stem/progenitor cells (HSPCs), are once again in the spotlight. Currently, striking evidence indicates that HSPCs and their residence tissue, the bone marrow (BM), are the main culprits in determining health and disease in the CV system. In this review, we place particular emphasis the role of EPCs and of their ancestors, HSPCs, in CV disease (CVD), shedding light on the evolving concept of the definition of EPCs. Finally, we provide a recent overview on the use of this cell population in clinical translational studies.

## 2. Lost in EPC Definitions

In 1997, Asahara et al. identified the putative EPCs in the mononuclear cell fraction of human peripheral blood, exploiting two antigens that are shared by endothelial and hematopoietic stem cells (HSCs), namely CD34 and VEGFR2 [1]. Since then, the research has strived to accurately identify EPCs to assure the most appropriate cell population in view of future clinical applications. Ideally, from a biological point of view, the EPCs should be endowed with an endothelial phenotype, self-renewal potential, and the capacity to differentiate into endothelial cells and form blood vessels in the living body [2]. Based on these definition criteria, two different approaches of studies have been used so far: (a) the identification of specific cell markers by flow cytometric assay of peripheral blood samples; and (b) the isolation of putative EPCs by cell culture methodologies.

The first approach, based on the use of monoclonal antibodies and flow cytometric analysis, allowed the enumeration of specific circulating cell subpopulations. On the assumption that endothelial and blood cells have a common embryological origin [3,4], several groups of laboratories have attempted to identify circulating EPCs by the CD34 marker in combination with endothelial antigens (i.e., VEGFR-2, CD146, CD144, CD31 Tie-1, Tie-2) [5,6]. In addition, as some mature endothelial cells may circulate in the bloodstream of healthy and diseased subjects [7,8], Peichev et al. introduced CD133 as a marker to discriminate EPCs from circulating mature endothelial cells (CECs) [9]. Despite the consensus statement that CD34^+^-CD133^+^-VEGFR-2^+^ expression should better identify circulating EPCs, and the current use of this nomenclature to correlate their concentration with severity and disease state, Case et al. demonstrated that it was not possible to distinguish immature EPCs by CD34, CD133, or VEGFR-2 expression because these surface markers also identified primitive HSCs [10]. Indeed, once isolated, they behaved as hematopoietic colony forming cells and did not form endothelial cell-lined vessels [10,11]. In conclusion, the phenotype that best represents circulating EPCs in terms of cell markers is still unknown.

The second approach was based on cell culture methodologies. Considering that circulating EPCs are an extremely rare cell population, researchers developed ex vivo isolation protocols based on the expansion of peripheral blood mononuclear cells (PBMCs) in pro-angiogenic media. Interestingly, despite the use of non-standardized culture protocols, different groups of laboratories achieved the same results, with only small differences. Indeed, Kalka and colleagues, and later Hur et al., identified two types of EPCs named “early” and “late” on the basis of their time of appearance in vitro [12,13]. Specifically, “early EPCs” were detectable after 7–10 days of PBMC culture in pro-angiogenic media on human fibronectin-coated dishes after the removal of non-adherent cells at day 4. The cells, characterized by clusters with spindle-shaped morphology, displayed limited proliferative potential and strong pro-angiogenic paracrine activity. Flow cytometric analysis revealed that the majority of the cells were composed of “hematopoietic angiogenic cells,” including monocyte/macrophages and lymphocytes (Figure 1). Three weeks after plating, cell colonies characterized by “cobblestone” morphology, high proliferative potential, null secretive activity, and endothelial cell lineage marker expression appeared in the same dish (Figure 1). These cells were named “late EPCs” or ECFCs (endothelial colony-forming cells). 

Similar results were also obtained by the laboratories of Lin and Ingram [14,15] with some modifications (Figure 2). Interestingly, preclinical studies demonstrated that both EPC types were equally able to restore the perfusion after hindlimb ischemia [13]. Overall, unlike the flow cytometric approach, the cell culture method of EPC isolation provided important biological information: (a) there is a crosstalk between myeloid and lymphoid cells and EPCs that may promote their expansion and differentiation [16], and (b) undoubtedly, late EPCs were derived from a rare heterogeneous PBMC population hidden in the early EPCs. This latter hypothesis was further corroborated by the finding that CD34^+^-depleted PBMCs, as well as cord blood-derived cells, were unable to generate any late EPC colonies, suggesting that late EPCs were raised from the CD34^+^ stem cell population [17]. In addition, by plating PBMCs on fibronectin-coated dishes in angiogenic growth medium for 48 h to avoid contamination by circulating endothelial cells and, after removing and seeding non-adherent cells into another fibronectin-coated dish for an additional 7 days, Hill’s group [18] obtained cell clusters morphologically similar to early EPCs, named endothelial cell colony-forming units (CFU-EC, also known as CFU-Hill), but not late EPCs, even after 3 weeks of culture (Figure 2). This evidence clearly indicated that EPC population resides in the discarded adherent cell fraction. Flow cytometric analysis showed that the clusters generated by Hill et al. were mainly composed of differentiated myeloid and lymphoid cell subsets that were unable to in vivo form blood vessels in experimental models [19]. The CFU-Hill colonies have been extensively used by clinicians as a predictive biomarker of vascular disease owing to their high correlation with the Framingham CV risk factor score. However, numerous authors equally and improperly grouped early EPCs and CFU-Hill culture techniques as the same method [2,20] (Figure 2). 

The culture approach firmly demonstrated that it was possible to in vitro obtain two groups of culture-committed cells characterized by intermediate stages of differentiation and different pro-angiogenic properties, namely instructive (release of angiogenic cytokines) and structural (vessel incorporation and stabilization). However, in this case, controversial findings have also accumulated over the years. Indeed, the development of numerous and different protocols for EPC enumeration and cultivation resulted in the identification of various EPC subsets that have been differently named according to their cellular origin, phenotype, and in vitro and in vivo properties, resulting in a general lack of consensus in the nomenclature (Figure 3) [2]. However, regardless of the multiple names used to identify the two different culture-derived cell populations, we should not underestimate that there is no demonstration of their existence in vivo and they may represent an in vitro artifact.

## 3. Circulating HSPCs and CV System: The Unifying Concept of EPCs

Notwithstanding the current impossibility to phenotypically separate EPCs from HSPCs because of their overlapping phenotype [21], EPCs have been universally considered as integrated components of the CV system committed to maintenance of endothelial integrity and vascular health. Consequently, their numerical reduction and dysfunction have been physiopathologically linked to the onset and progression of CVD in numerous clinical conditions, transforming this cell population in a powerful predictive biomarker of CVD [22,23,24]. During clinical studies aimed at circulating CD34^+^-VEGFR-2^+^ EPC enumeration, it was also noted that circulating CD34^+^ cells were reduced in CVD condition. In this regard, the literature provides compelling evidence that, unlike CD34^+^-VEGFR-2^+^ EPCs, circulating CD34^+^ cells as well as CD34^+^-CD133^+^ cells significantly predicted disease severity, mortality, and improved risk stratification in subjects with different CVDs and other CV-related disorders [25,26,27,28,29,30,31]. In 2016, these observations were further supported with a meta-analysis by Rigato et al., which showed that circulating CD34^+^-CD133^+^ cells were the most predictive phenotype for CV events, restenosis after endovascular intervention, CV death, and all-cause mortality [32]. One year later, Fadini et al. demonstrated that in patients with type 2 diabetes (T2D), the reduced baseline levels of circulating CD34^+^ stem cells were a long-time (up to 6 years) predictive biomarker of adverse CV outcomes [33]. However, almost simultaneously Hayek et al. [34] reported that low CD34^+^-VEGFR-2^+^ cells, but not CD34^+^ cells, predicted the risk of mortality and peripheral artery disease (PAD)-related events. These discrepancies may be ascribable to technical reasons, such as lot-to-lot variability of anti-VEGFR-2 antibodies that, unlike anti-CD34^+^ antibodies, are not clinical grade but for research use only, and to the different flow cytometry gating strategies used for event acquisition. Nevertheless, taken together, these data provide important insights: (i) the most predictive circulating stem cell population could be disease-specific [35]; (ii) CD34^+^-VEGFR-2^+^ EPCs are progeny of HSPCs and are rarer than expected; and (iii) circulating CD34^+^ stem cells are mainly composed by HSPCs and are endowed with vasculotrophic properties. 

This latter assumption is corroborated by clinical evidence that circulating HSPCs increase by 25% in patients after acute myocardial infarction (AMI). This mobilization, which starts within few minutes and persists for several days after AMI, is suggestive of the active participation of this cell population in CV repair [36,37,38] as, consistently, its mobilization failure correlates with poorer prognosis [39]. In addition, classical CV risk factors such as smoking, hypertension, dyslipidemia, diabetes, obesity, and aging are associated with a decline in circulating HSPCs number and function [40,41], supporting the concept that CVD is the result of an “impaired damage control” condition, wherein tissue damage is not adequately counterbalanced by endogenous repair [42,43]. The mechanistic concept of CV defective repair was supported by the observation that patients with extreme-duration type 1 diabetes without CVD had preserved levels of stem/progenitor cells, suggesting a direct involvement of these cells in the neutralization of adverse effects promoted by the metabolic abnormalities on vascular tissue [44]. However, there is no direct demonstration of whether therapeutic strategies aimed at restoring or increasing CD34^+^ stem cell number and function can also exert CV protective effects. 

Finally, CD34^+^ stem cells demonstrated their vascular regenerative capacity and proangiogenic potential as successfully used in regenerative medicine for the treatment of limb and cardiac ischemia [36,45]. Collectively, this experimental evidence suggested an interdependent relationship among HSPCs, the CV system, and the BM, because the origin of circulating CD34^+^ stem cells must be tracked back in the BM, and led to the reassessment of EPC concept in terms of definition, identity, origin, and function at CV level. The emerging concept is that the whole circulating CD34^+^ stem cell population, of which EPC progeny is a minor subset, is the true executor of CV health maintenance.

## 4. Intertwined Relationship of Circulating CD34^+^ Progenitor Cells with the Cardiovascular System

As previously detailed, circulating CD34^+^ HSPCs comprise several subsets of CD34^+^ cells, including EPCs, (CD34^+^CD133^+^/KDR^+^) that can actively participate in vascular repair and growth [46,47]. Besides reflecting vascular integrity and having been used as biomarkers of vascular repair [48,49], their low CD34^+^ HSPC circulating number in individuals with CVD (e.g., heart failure or acute coronary syndromes) predicts higher mortality risk [50] strengthening the impact of their dysfunction on endogenous vascular repair capacity. A number of studies has shown that lifestyle behaviors and environment can significantly affect CV health. To this regard, obesity [51], smoking [52], and physical inactivity [53] are associated with increased CVD and a greater risk of mortality. Exercise training is known to improve vascular function and this effect seems to be, in part, mediated by the modulation of circulating CD34^+^ cell number and function [54]. Indeed, numerous studies have demonstrated that chronic and acute exercise both improve cardiovascular health and promotes the mobilization of CD34^+^ cell from the BM to peripheral blood compartment where they realistically exert their vasculotrophic functions [55,56,57,58,59]. However, if we can beneficially affect our CV system health by improving our lifestyle, particular attention should be paid to the environment where we live. It has been demonstrated that fine particulate matter (PM) exposure, consisting of mixture of various particles including crustal material, metals, and bioaerosols, can induce cardiovascular disorders such as reduced heart rate variability, vascular dysfunction, and enhanced coagulation–thrombosis [60,61]. Again, these effects were associated with a significant suppression of circulating CD34^+^ stem cell number and function in both mice and humans with mechanisms involving PM-mediated ROS production [62,63,64,65].

In recent years, investigations have been devoted to understanding the significance of circulating CD34^+^ proangiogenic progenitors (EPCs) in host defense during sepsis-induced vascular injury.

Extensive endothelial cell damage, especially in the microcirculation, frequently occurs during sepsis. Multiple pathological factors, including systemic inflammation, oxidative stress, toxic compounds generated by invading microorganisms, and unbalanced release of vasoactive mediators synergistically lead to the loss of endothelial barrier function resulting in tissue edema, the collapse of circulation, and final vital organ system failure (multiple organ dysfunction syndrome caused by sepsis, MODS). The host organism reacts to massive endothelial damage through BM mobilization and the activation of resident progenitors that are part of the intrinsic mechanisms involved in the maintenance of vascular homeostasis and regeneration [66]. Numerous studies have repeatedly reported the increase in circulating CD34^+^ progenitors as well as their active recruitment and homing to tissue sites of injury in response to septic challenges, both in pediatric and adult patients [67,68,69]. The recruited vascular progenitors can then participate in the host repair of damaged endothelium via direct endothelial differentiation and/or the release of angiogenic mediators and extracellular vesicles (EVs) [70,71,72]. To this regard, EVs, not replicating lipid bilayer particles released from the cell, have been shown to have an important role in the intercellular communication of circulating CD34^+^ progenitors by which they can mediate and modulate pro-angiogenic/vasculogenic and anti-inflammatory paracrine effects [73,74]. In recent years, the striking significance of CD34^+^ vascular progenitors in host defense has promoted numerous lines of investigation for the development therapeutic strategies in sepsis-induced vascular injury, mainly based on CD34^+^ progenitor transplantation or EV delivery [66].

### Circulating CD34^+^ Progenitor Cells and Cancer

Tumor growth and progression require nutrients and oxygen to proliferate. These physiological tumor needs are achieved by new blood vessel formation, of which the network in turn facilitates cancer metastasis into the systemic circulation. Accumulating data indicate that circulating vascular progenitors not only provide structural support to nascent vessels [75,76] but also actively participate in new blood vessel formation via the paracrine secretion of pro-angiogenic factors [73,74]. The CD34^+^ progenitor cell-mediated neovascularization process involves multiple steps, including BM cell mobilization, recruitment, and homing to neovessel sites, which are finely orchestrated by tumor cells [77]. A wide range of cytokines and chemokines, such as VEGF and SDF-1α, released by tumor microenvironment promote vasculogenesis and cancer progression by mobilization of BM resident CD34^+^ progenitors in the systemic circulation and enhancement of their recruitment to the tumor site [78,79,80,81]. The inhibition of CD34^+^ progenitor-mediated vasculogenesis, likewise anti-angiogenic therapies, showed significant reduction in tumor neovasculogenesis and development [82,83]. A comprehensive understanding of the molecular mechanisms subtending CD34^+^ cell-mediated neovascularization may provide novel therapeutic strategies in cancer treatment.

## 5. The CV Tropism of HSPCs: Origin, Role, and Pathogenic Implications in CVD

HSPCs are commonly defined by CD34 expression. CD34 is a transmembrane adhesion phosphoglycoprotein, with unknown function. First identified on HSPCs, its expression has been uniquely related to this cell population because clinically associated with selection of stem cells with hematopoietic properties [84]. Conversely, from a prevailing school of thought, strong evidence demonstrates that CD34 is expressed by a numerous other nonhematopoietic cell types, including mesenchymal stem cells (MSCs), interstitial cells, epithelial cells, keratocytes, and endothelial cells where it represents a small proportion of the total cell population and defines a distinct cell subset with progenitor activity [85,86,87,88]. It is universally recognized that stem cells reside in adult tissues. This is especially true of the vascular system, where resident and circulating HSPCs play an active role both in normal homeostasis maintenance and disease progression [89].

What is the relationship between HSPCs and vascular cells? On the basis of their close kinship and proximity during early development, Florence Sabin in 1920 [Contrib. Embryol. Carnegie Inst. 9, 215–262 (1920)] supposed the existence of the hemangioblast, a putative common progenitor of hematopoietic and endothelial cells [90]. Although the existence of the hemangioblast is still debated 100 years later, current data support the concept of the “hemogenic endothelium” according to which terminally differentiated endothelial cells generate hematopoietic cells during a specific stage of embryonic development [91,92]. Specifically, HSPCs arise from a region of the mammalian embryo encompassing the aorta, gonads, and mesonephros (AGM) from which they detach into circulation, reach the liver, and finally colonize the BM. There, inside the hematopoietic BM niche, a functional structure devoted to their quiescence, differentiation, and egression maintenance, they generate all cells of the blood lineage throughout life span [93,94]. At the molecular level, the endothelial hematopoietic transition envisages the activation of the transcription factor RUNX1 (runt-related transcription factor 1) that is normally downregulated by HOXA3 (Homeobox A3) in endothelial cells. RUNX1 expression is then suppressed once HSPCs are formed [95,96]. The hemato–vascular overlap observed during embryological development can thus explain the strong vascular tropism displayed by HSPCs.

For decades, scientists considered HSPCs as a static source of stem cells indefinitely located in the BM with the unique role of replenishing blood and immune system cells only when called upon. However, HSPCs are also present in the peripheral blood (PB) where they represent the 0.05% of total white cells [97]. Although it was believed that this small amount was a passive leak from the BM, today it is clear that HSPCs are actively released from the BM to PB from which they reach various organs and eventually come back home into BM in few hours in accordance with circadian rhythm [98,99,100]. But the question now is: Why? What exactly are they doing? Although today the biological meaning of HSPCs in–out is not completely understood yet, it is assumed that the continuous in-out trafficking has various role: (i) to patrol peripheral tissue contributing to local immunosurveillance and inflammation [98]; (ii) to allow a better redistribution and replenishment of BM niche improving normal hematopoiesis [99]; and finally (iii) to contribute to the maintenance of tissue homeostasis by regulating and promoting endogenous repair [98,101,102]. This latter assumption was supported by compelling evidence that donor cells can repopulate nonhematopoietic tissue such as the lungs [103], liver [104], kidney [105], and including myocardium [106] and endothelium [107] in BM-transplanted individuals. In this regard, it is noteworthy to mention the study of Jiang et al., who demonstrated that 2% of endothelial cells in the skin and gut of patients with hematologic malignancy who underwent BM CD34^+^ cell transplantation were of donor origin [108]. Similarly, Peters et al. demonstrated by in situ hybridization with sex chromosome-specific probes that 4.9% of blood vessel in the histological samples of various cancers originated from BM donors of the opposite sex [109]. Although recent literature also shows the existence of vascular and myocardial niches as alternative source of vasculotrophic progenitors [110,111,112,113]; taken together, these data suggest that progenitor cells with pro-angiogenic properties represent a small CD34 subset and that the number and functionality of circulating HSPC population, as whole, is the mirror and at the same time the determinant of the cardiovascular and general health.

### Clinical Implications of HSPC Dysfunction

In view of the aforementioned role of circulating HSPCs, it is clear that their numerical and functional perturbation implies profound and severe clinical sequelae, including increased mortality [114]. So far, the dysfunctions of circulating HSPCs have mostly been studied and interpreted in view of their pro-angiogenic counterpart, namely the EPC progeny. Countless studies have mechanistically detailed the alterations promoted by different pathological contexts on the main EPC functional processes, including BM mobilization, migration, homing, and differentiation [10,75,115], overlooking that CD34^+^ cells, as HSPCs, are sources not only of pro-angiogenic cells but also of immune system cells. The profound link existing between HSPCs and cardiovascular system was made more evident by the clonal hematopoiesis of indeterminate potential (CHIP) a clinical condition in which the mutation in only handful genes, specifically DNMT3A (DNA MethylTransferase 3 Alpha), TET2 (ten eleven translocation (Tet) methylcytosine dioxygenase), AXL1 (AneXeLekto receptor tyrosine kinase), PPM1D (Protein Phosphatase, Mg^2+^/Mn^2+^ Dependent 1D), JAK2 (Janus Kinase 2), TP53 (Tumor Protein p53), SF3B1 (Splicing Factor 3b Subunit 1), and SRSf2 (Serine/arginine-Rich Splicing factor 2) in HSPCs promotes the accumulation of clones of mutated leukocytes that populate the PB. This pre-malignant state, although considered as the first step towards leukemia, rarely develops malignancy (only 0.5% to 1% per year, hence the term ”indeterminate potential”), but represents a potent and independent cardiovascular risk factor (40% increase) [116]. Concerning the mechanisms proposed to link CHIP and cardiovascular events, it has been observed that in mice engineered to bear mutations in genes commonly involved in CHIP (e.g., Tet2), there was an increased expression of proinflammatory mediators implicated in the pathogenesis of atherosclerosis, such as cytokines interleukin (IL)-1b and IL-6 [117]. Similarly, atherosclerosis, hypercholesterolemia, hypertension, and diabetes are associated with the elevation of hematopoiesis with a myeloid bias that suggests a contribution of inflammatory leukocytes to the development and progression of CVD. In this regard, Terenzi et al. demonstrated, by a multiparametric flow cytometry assay, profound differences in circulating proangiogenic and proinflammatory progenitor cell content between patients with T2D and age-matched control subjects [118]. Specifically, patients with T2D displayed an increased frequency of proinflammatory myeloid cells and decreased frequency of circulating monocytes with an M2 phenotype, which is associated with proangiogenic and anti-inflammatory functions, and a reduction of proangiogenic CD34^+^ progenitor cells with primitive (CD133) and migratory (CXCR4) phenotypes. The flow cytometric assessment of the balance between circulating vascular regenerative progenitor cells and inflammatory cells in patients with T2D could represent a promising translational approach for identifying patients with T2D at increased risk for cardiovascular comorbidities [118]. Interestingly, preclinical and clinical evidence showed that different pathological milieus, including diabetes, are able to redirect HSPC differentiation toward pro-inflammatory and harmful cell populations with pro-calcific and profibrotic properties [119,120,121,122,123,124] by molecular and epigenetic mechanisms that could already take place at the BM level (Figure 4) [125].

## 6. BM as Master Regulator of Global Organismal Health

A complex signaling network triggered by internal, metabolic, extrinsic factors (outside the bone marrow) and finely orchestrated in a physically defined area of BM, named the HSPC niche, regulates, by epigenetic modifications, the differentiation and maintenance of HSPCs pool, granting an adequate blood cell production, both under steady-state and stress conditions [126]. At the BM level, the perturbation of this delicate balance is associated with adverse clinical sequelae, including excess mortality [114]. Indeed, although the links between hematological disturbances and accelerated CVD remain unclear, both BM failure and alteration in circulating HSPC level and phenotype are associated with clinical conditions characterized by premature death [127]. These observations increased the interest in the BM status of patients with CVD or other risk factors [128,129,130] transforming the role of this organ from merely dedicated to the generation of blood elements, to central housekeeper of global organismal health [114]. Interestingly, experimental studies in animal models of atherosclerosis and diabetes have revealed the reprogramming of HSPCs in the BM as a cause of changes in circulating innate immune cells that cause predisposition to CVD. Hyperlipidemia showed the promotion of inflammatory activation of monocytes by epigenetic reprogramming of HSPCs in low-density lipo- protein receptor deficient (Ldlr−/−) mice [131]. In a mouse model of diabetes, epigenetic modifications were responsible for the activation of inflammatory genes in BM progenitors that persisted in the differentiated progeny [132,133,134,135]. Similar mechanisms seem to be involved in the monocytosis and neutrophilia of humans with chronic stress [136]. 

However, such data do not provide any clue whether the differentiation drift of HSPCs has to be considered as a mechanistic factor of disease or an epiphenomenon of BM dysfunction.

## 7. Vascular Progenitors: Are All They of BM Origin?

In addition to BM stem and progenitor cells with an embryonic developmental origin, the existence of a class of vascular progenitor cells in the adult vasculature has been debated. Within mural cells, vascular pericytes have reported to possess stem cell properties in adult tissue [137]. These cells, identified by several markers including smooth muscle α-actin (SMA), desmin, NG-2, and platelet-derived growth factor receptor (PDGFR)-β, can be expanded clonally in vitro and possess properties of mesenchymal stem cells. They have the potential to give rise to different tissues in vitro, but this is not clear in vivo. Mounting evidence suggests a model in which mesenchymal stem cells remain entangled in developing vessels and become pericytes. These cells, which maintain stem cell properties, can be reactivated after injury and disease, providing instructive and structural information during vascular regeneration. However, as their identifying markers are not restricted to pericytes, the contributions from cells other than pericytes cannot be ruled out. The presence in vessels of pericytes with mesenchymal stem cell properties and of other vascular stem or progenitor cells could have important therapeutic implications as they could be harvested for tissue regeneration [138]. 

## 8. Clinical Use of CD34^+^ Cells: How Far Are We from Using Them?

The regenerative potential of BM-derived stem cells (SC) and their capability to differentiate into a wide-ranging set of cellular types in vitro and in animal models [139] piqued the interest of researchers assessing their possible role in clinical practice. Most of the initial cell therapy studies were based on the administration of a whole unfractionated BM-SC population, containing HSPCs, mesenchymal stem cells, and EPCs, usually obtained from iliac crest aspiration. Nevertheless, CD34^+^ cells were rapidly recognized as the most relevant subpopulation in terms of bioactivity and therefore rapidly became an appealing instrument for researchers, despite a more complex selection procedure and a lower yield in number of cells (they typically represent only 0.5% to 6% of unfractionated BMCs) [140,141]. The most relevant clinical trials involving the administration of CD34^+^ cells were designed to evaluate their therapeutic potential in various CV and non-CV disease settings, such as refractory angina (RA), left ventricular systolic dysfunction (LVSD), liver failure, and complications of type I diabetes.

### 8.1. Refractory Angina

A relevant number of preclinical studies has addressed the need to find new therapeutic options for RA, a condition defined as “symptoms due to established reversible ischemia in the presence of obstructive coronary artery disease (CAD), which cannot be controlled by escalating medical therapy with the use of second- and third-line pharmacological agents, bypass grafting, or stenting including percutaneous coronary intervention (PCI) of chronic total coronary occlusion”. RA is a main CV problem, affecting about 5–10% of chronic stable CAD patients and is responsible for (i) frequent hospitalizations, (ii) a high amount of resource utilization, and (iii) a poor quality of life for CAD patients [142,143]. It was already known from these studies that, in the rat, intramyocardially transplanted CD34^+^ cells after myocardial infarction could contribute to cardiomyogenesis and vasculogenesis, probably by both a direct regenerative effect and a paracrine secretion of growth factors and cytokines [144]. On this basis, Losordo et al. assessed, for the first time, the safety and bioactivity of intramyocardial administration of CD34^+^ cells in 24 patients with RA, in a phase I/IIa, double-blind, randomized controlled trial (RCT, Table 1). All patients received granulocyte-colony stimulating factor (G-CSF) for mobilization, followed by leukapheresis for collection of mononuclear cells. Then, CD34^+^ cells were purified and intramyocardially injected during NOGA electromechanical mapping in treatment group patients, which were divided in three dose cohorts (5 × 10^4^, 1 × 10^5^, and 5 × 10^5^ CD34^+^ cells/kg), while placebo group patients received an intramyocardial injection of cell diluent. Despite RA transiently worsening in 13 patients after G-CSF administration, probably due to its intrinsic side effects of increased blood viscosity, metabolic demand, and platelet count, and one placebo group patient developed ventricular tachycardia, successfully cardioverted, during mapping procedure, the protocol was judged adequately safe overall. The efficacy endpoints included angina frequency, nitroglycerine usage, exercise tolerance, Canadian Cardiovascular Society (CCS) class, and Seattle Angina Questionnaire assessment; despite a global amelioration in all treatment and placebo groups at 6 months, probably due to an overall strong placebo effect of the procedure, treatment groups showed a better trend for all of efficacy endpoints, with no significant difference amongst the three different doses groups, although the study was not really powered to detect it [145].

Similar results came from a subsequent double-blind, multicentric, controlled phase II RCT, the ACT34-CMI in which 167 patients were enrolled (Table 1). Indeed, patients who received intramyocardial administration of autologous CD34^+^ cells (1 × 10^5^ or 5 × 10^5^) experienced a significant amelioration of angina frequency and exercise tolerance at 6 and 12 months from injection, and a reduction in mortality rate at 12 months (i.e., 5.4% in the placebo group vs no deaths in the treatment group) [111]. A significant reduction in angina frequency and a trend towards lower mortality were confirmed at 24 months follow-up for both treatment groups [36]. Unfortunately, the phase III RENEW study, which was supposed to include 444 patients, was prematurely terminated by the sponsor after the enrolment of only 112 and thus, despite the results were consistent with previous studies, it was underpowered to provide conclusive evidence of efficacy and safety of CD34^+^ cells intramyocardial administration (Table 1) [146,147]. Current (2019) European Society of Cardiology (ESC) Guidelines for the diagnosis and management of chronic coronary syndromes, based on the results of a pooled analysis of the three aforementioned RCTs that substantially confirms a positive effect of CD34^+^ intramyocardial injection on exercise time and angina frequency, classify this procedure amongst the “potential treatment options for refractory angina”, but underscore the need for larger RCTs in order to elaborate a clear recommendation [142].

### 8.2. Left Ventricular Systolic Dysfunction

Most of the CD34^+^ studies from Vrtovec’s group addressed non-ischemic dilated cardiomyopathy (DCM). A first, pilot, open-label clinical study included 55 patients (i.e., 28 treated, 27 controls) with a diagnosis of DCM with a left ventricular ejection fraction (LVEF) < 30% and a New York Heart Association (NYHA) Class ≥ 3 for at least 3 months before referral (Table 2). Following mobilization by G-CSF and apheresis, CD34^+^ cells were injected in the coronary artery supplying the less viable segments (localized by myocardial scintigraphy). After 1 year, LVEF and 6-minute walk distance performance significantly increased in the treatment group, while N-terminal pro-brain natriuretic peptide (NT-proBNP) values and a composite endpoint of mortality or heart transplantation was significantly lower [148]. Similar results came from another study evaluating the 5-year follow-up in 110 (i.e., 55 treated, 55 controls) after treatment. Moreover, a higher CD34^+^ cell engraftment correlated with a better treatment response [149]. Good results have been obtained also for ischemic cardiomyopathy (ICM) after an intramyocardial injection of cells in the hibernating myocardium (Table 2) [150]. The intramyocardial injection seems to be more effective than intracoronary in the DCM form as well [151]. Moreover, there is evidence that right ventricular function could also benefit from left ventricle intramyocardial administration of CD34^+^ cells, most likely due to the functional interdependence between the left and right sections of the heart [152].

### 8.3. Use of CD34^+^ in Conditions Other Than CVD

Moving to a non-CV condition, CD34^+^ cells have been demonstrated to be capable to differentiate into hepatocyte as well. Gordon and colleagues assumed that the regenerative effect observed in previous studies involving other BMC populations [153,154,155] had to be ascribed to CD34^+^ cells; therefore, they attempted intraportal or via hepatic artery infusion of CD34^+^ autologous cells after mobilization with G-CSF, in 3 and 2 patients with liver insufficiency, respectively (Table 3). No serious complications were observed. Interestingly, three out of five patients improved in terms of serum bilirubin and four out of five in terms of serum albumin at 60 days [153]. Bilirubin levels slightly increased back after 12-18 months follow up, but without the detection of long-term complications [153]. The same research group later obtained further encouraging results from hepatic artery infusion of CD34^+^ cells in 9 patients [156]. Moreover, Sharma et al. compared hepatic artery infusion of autologous CD34^+^ cells with standard of care in 45 non-viral cirrhosis patients (i.e., 22 in the treatment group, 23 in the control group) and observed a significant improved liver function after cell therapy, as reported in Table 3 [157]. Regenerative potential is not the only clinically relevant feature of CD34^+^ cells: they also possess immunoregulatory properties, as shown in multiple sclerosis studies [158]. Some clinical trials enrolled patients with early onset type I diabetes mellitus, who underwent the so called nonmyeloablative hematopoietic stem cell transplantation (AHST) of autologous CD34^+^ cells. Beneficial effects of this treatment have been observed in terms of autoimmunity, β-cell function, and insulin dependency, with some patients remaining insulin independent for several years (Table 3) [159,160,161].

## 9. Conclusions

In the past, researchers have been focused on the quantification and identification of the circulating “true EPCs”, while today the interest is gradually shifting to the study of the complex network existing among BM, circulating HSPCs, and the CV system. Over the years, the numerous theories regarding EPC origin led to opposing schools of thought that, from a holistic point of view, were not mutually exclusive anyway. The intimate relationship and legacy existing between hematopoietic and CV system are witnessed by the emerging cross-talk signals that, through the BM, take part in cardiac and vascular regulation, both in health and disease conditions. In the clinical arena, the quantification of circulating HSPC subpopulations, not limited to the vasculotrophic phenotype, could provide additional physiological and prognostic information improving CV risk stratification throughout patient lifespan. In addition, a better understanding of cellular and molecular mechanisms subtending BM dysfunction and BM stem/progenitor cells could boost the development of novel therapies for CVD.

## Figures and Tables

**Figure 1 cells-12-00112-f001:**
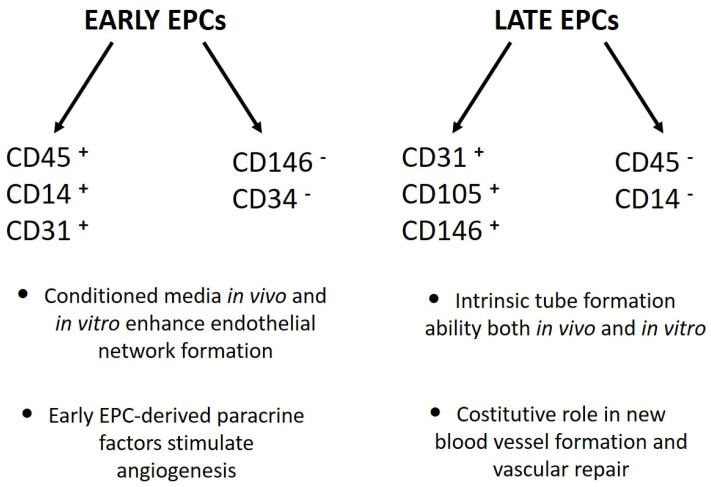
Phenotypical and functional differences between “early” and “late” EPCs [12,13,14].

**Figure 2 cells-12-00112-f002:**
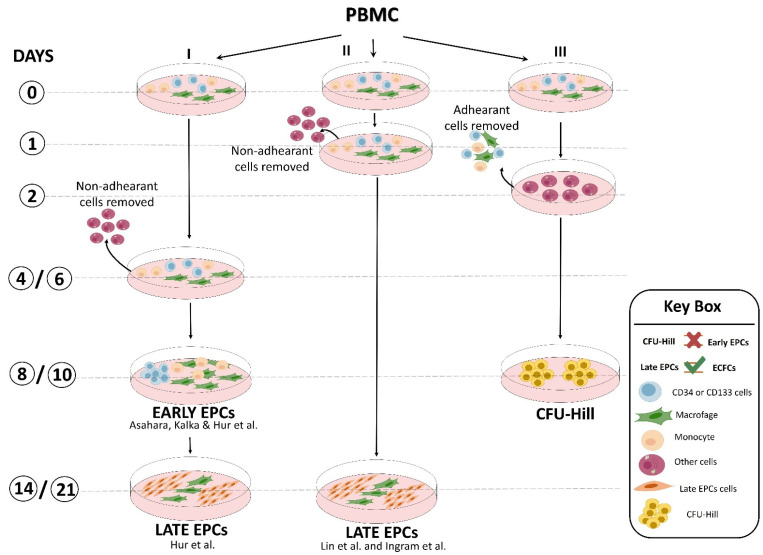
Representative scheme of culture methods developed by different laboratories to obtain “early EPCs”, “late EPCs” (named also ECFCs; endothelial colony-forming cells), and CFU-Hill [12,13,14,15].

**Figure 3 cells-12-00112-f003:**
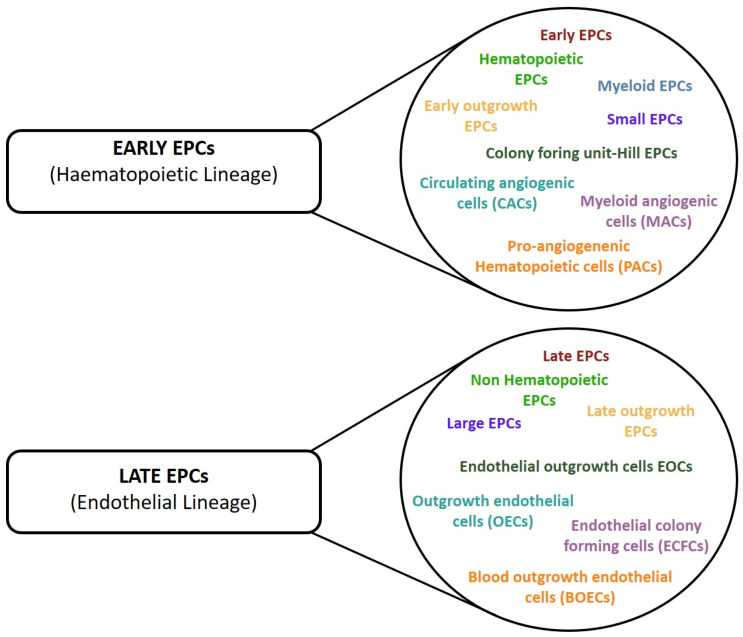
Different nomenclature used in the studies to define cells with pro-angiogenic properties. The cells have been divided into two group according to their lineage phenotype.

**Figure 4 cells-12-00112-f004:**
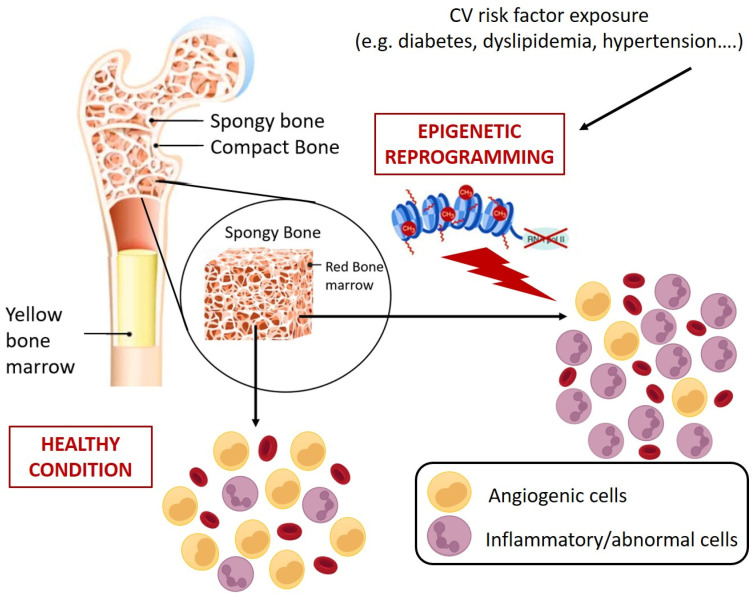
Schematic representation of CV risk-induced reprogramming of HSPCs at bone marrow level. CV risks promote epigenetic changes in HSPCs that results in abnormal expansion of cells with inflammatory and pro-atherosclerotic features. Specifically, monocytes characterized by more inflammatory phenotypes (e.g., alternative and non-classic monocytes) and generation of circulating progenitors with pro-calcific and pro-fibrotic characteristics [119,120,121,122,123,124].

**Table 1 cells-12-00112-t001:** Clinical trials on refractory angina.

Condition	Population	Study Design	Outcomes	Major Adverse Events	Ref
**Refractory Angina**	24 pts (>21 yrs old), CCS class III or IV angina, on OMT, not amenable to revascularization	Phase I/IIa, double-blind, RCT. Intramyocardial injection of autologous CD34^+^ cells (5 × 10^4^, 1 × 10^5^, or 5 × 10^5^ cells/kg vs cell diluent)	Treatment better than placebo in angina frequency, nitroglycerine usage, exercise tolerance, CCS class, and Seattle Angina Questionnaire assessment. No dose–response observed.	1 pt in ctr group developed ventricular tachycardia during mapping procedure, then successfully cardioverted.	[145]
	167 pts (21–80 yrs old), CCS class III or IV angina, on OMT, not amenable to revascularization	Phase II, double-blind, RCT. Intramyocardial injection of autologous CD34^+^ cells (1 × 10^5^ or 5 × 10^5^ cells/kg vs cell diluent)	Low-dose treatment significantly better than placebo in angina frequency and exercise tolerance at 6 and 12 mo. Both low- and high-dose better than placebo in angina frequency at 24 mo.	MACE/deaths 33.9/12.5% in ctrl group, 21.8/1.8% in low-dose group and 16.2/3.6% in high-dose group at 24 mo.	[111]
	112 pts (18–80 yrs old), CCS class III or IV angina, on OMT, not amenable to revascularization, LVEF ≥ 25%, reproducible exercise-limiting angina	Phase III, double-blind, RCT. Intramyocardial injection of autologous CD34^+^ cells (1 × 10^5^ up to 1 × 10^7^ cells/kg vs cell diluent (active control) vs standard of care control)	Change in TET in treated pts vs placebo was 42.1/61.0 s in ITT/PP populations at 3 mo, 34.7/46.2 s at 6 mo, and 20.4/36.6 s at 12 mo. Angina RR in treated pts vs placebo was 0.77/0.80 in ITT/PP at 3 mo, 0.58/0.63 at 6 mo, and 1.02/0.95 at 12 mo.	MACE/deaths 67.9/7.1% in standard of care ctrl group, 42,9/10.7% in active ctrl group, and 46.0/4.0 in treated group at 24 mo.	[146,147]

Abbreviations: CCS = Canadian Cardiovascular Society; ITT = intention to treat; LVEF = left ventricular ejection fraction; MACE = major adverse cardiovascular event; OMT = optimal medical therapy; PP = per protocol; PTs = patients; RCT = randomized controlled trial; RR = relative risk; TET = total exercise time.

**Table 2 cells-12-00112-t002:** Clinical trials on dilated and ischemic cardiomyopathy.

Condition	Population	Study Design	Outcomes	Major Adverse Events	Ref
**DCM**	55 pts (>18 yrs old), DCM, LVEF < 30%, NYHA Class ≥ III for 3 mo before referral, on OMT for at least 6 mo.	Phase II, open-label, RCT. Intracoronary infusion of autologous CD34^+^ cells (average 123 ± 23 × 10^6^) vs no infusion.	Significant improvement in LVEF and 6 MWT distance in treated pts vs ctrl group at 3 and 12 mo. Significant reduction in cardiac mortality/heart transplantation in treated pts vs ctrl group at 12 mo.	No major procedure-related complications. No effects of transplanted stem cells on QTc interval and QT interval variability.	[148]
	110 pts (18–65 yrs old), DCM, LVEF < 30%, NYHA Class III for 3 mo before referral, on OMT for at least 6 mo.	Phase II, open-label, RCT. Intracoronary infusion of autologous CD34^+^ cells (average 113 ± 26 × 10^6^) vs no infusion.	Significant improvement in LVEF and 6 MWT distance in treated patients vs control group at 60 months. Significant reduction in total mortality in treated patients vs control group at 60 months (14% vs 35%).	No major procedure-related complications.	[149]
**ICM**	33 pts (>18 yrs old), ICM, no amenable to revascularization, LVEF < 40%, NYHA Class III for 3 mo before referral.	Phase II crossover study. In phase I, patients were treated for 6 mo with OMT. Thereafter, they crossed over to phase II where they received intramyocardial injection of autologous CD34^+^ cells (average 90.6 ± 7.5 × 10^6^).	No significant improvement in LVEF and 6MWT distance after OMT, significant improvement 6 mo after cell injection.	2 deaths during phase I	[150]

Abbreviations: 6MWT= 6-Minute Walking Test; DCM = dilated cardiomyopathy; ICM = ischemic cardiomyopathy; LVEF = left ventricular ejection fraction; NYHA = New York Heart Association; OMT = optimal medical therapy; RCT = randomized controlled trial.

**Table 3 cells-12-00112-t003:** Clinical trials on liver insufficiency and T1D.

Condition	Population	Study Design	Outcomes	Major Adverse Events	Ref
**Liver Insufficiency**	5 pts (20–65 yrs old), chronic liver failure, abnormal serum albumin and/or bilirubin and/or pro-thrombin time, unsuitable for liver transplantation, WHO performance status < 2.	Phase I clinical trial. 3 pts received 1 × 10^6^–2 × 10^8^ autologous CD34^+^ cells via portal vein, 2 pts received 1 × 10^6^–2 × 10^8^ autologous CD34^+^ cells via hepatic artery.	Improvement in serum bilirubin in 3/5 pts at 60 days maintained by only 1 patient at 12 mo. Improvement in serum albumin in 4/5 pts at 60 days maintained at 12–18 mo.	No major procedure-related complications.	[153]
	9 pts (20–65 yrs old), chronic alcoholic liver failure, abnormal serum albumin and/or bilirubin and/or pro-thrombin time, unsuitable for liver transplantation, WHO performance status < 2.	Phase II clinical trial. Injection of autologous CD34^+^ cells (average 229.7 × 10^6^) via hepatic artery.	Improvement in serum bilirubin at 12 weeks, transient improvement in ALT and AST. Improvement of Child–Pugh score in 7/9 patients and improvement of ascites in 5/9 patients at 12 weeks.	No major procedure-related complications.	[156]
	55 pts (18–70 yrs old), non-viral hepatic cirrhosis, MELD score > 14, requiring liver transplantation.	Phase II open-label non-randomized controlled CT. 22 patients unwilling for liver transplantation received autologous CD34^+^ cells via hepatic artery. 23 patients opted for regular inclusion in the institutional liver transplantation waiting list.	Transient improvement in serum albumin in treated pts (not sustained at 3 mo); improvement of serum creatinine and MELD score at 3 mo.	3 deaths in ctrl group (2 due to sepsis, 1 to gastrointestinal bleeding), 1 death in treatment group on 88th day after CD34^+^ cell infusion (due to sepsis). No major procedure-related complications.	[157]
**T1D**	23 pts (12–35 yrs old), diagnosis of T1D within the previous 6 weeks.	Phase I/II clinical trial. Pts underwent immune ablation with cyclophosphamide and ATG, followed by infusion via peripheral vein of autologous CD34^+^ cells (10.52 × 10^6^ cells/kg).	Most pts showed a reduction in Hb1Ac levels and an increase in C-peptide levels after treatment. 20 pts experienced time free from insulin (12 until the end of follow up, up to 4 yrs).	Bilateral nosocomial pneumonia (2 pts), posttransplant oligospermia (9 pts), Graves’ disease (1 pt), transient hypergonadotropic hypogonadism (1 pt), autoimmune hypoth-roidism (1 pt).	[159]
	24 pts (12–35 yrs old), diagnosis of T1D within the previous 6 weeks, sustained endogenous secretion of insulin and WHO performance status ≤ 2.	Phase II clinical trial. Pts underwent immune ablation with cyclophosphamide and ATG, followed by infusion via peripheral vein of autologous CD34^+^ cells (4.19 × 10^6^ cells/kg).	General reduction in Hb1Ac levels and increase in C-peptide levels after treatment. 20 pts experienced time free from insulin (4 until the end of follow-up, up to 80 mo).	ATG–related skin reaction/vasculitis (4 pts), neutropenic fever (12 pts), sepsis (4 pts, out of which 1 was fatal).	[160]
	40 pts (14–27 yrs old), recent diagnosis of T1D with time from symptom onset to AHST 4–26 weeks	Phase II, parallel-assignment, non-randomized clinical trial. Treatment group pts underwent immune ablation with cyclophosphamide and ATG, followed by infusion via peripheral vein of autologous CD34^+^ cells. Ctrl group pts received regular insulin therapy.	Increase in C-peptide levels in treatment group and decline in ctrl group at 48 mo. Comparable reduction in Hb1Ac levels in both groups. 14 pts in treatment group experienced time free from insulin (3 until the end of follow up, up to 48 mo). One pt in ctrl group experienced transient insulin independence for 7 mo.	Graves’ disease (2 pts on treatment, 1 pt in ctrl group), autoimmune thyroid disease (2 pts in ctrl group).	[161]

Abbreviations: ASHT = allogeneic hematopoietic stem cell transplantation; ALT = alanine aminotransferase; AST = aspartate aminotransferase; ATG = anti-thymocyte globulin; CT = clinical trial; Hb1Ac = glycated hemoglobin; MELD = model for end-stage liver disease; PT = patient; T1D = type 1 diabetes; WHO = World Health Organization.

## Data Availability

Not applicable.

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
