# Peer review of "The Long Telling Story of “Endothelial Progenitor Cells”: Where Are We at Now?"

_cells, 2022, doi:10.3390/cells12010112_

Round 1

Reviewer 1 Report

This article has an interesting title.  However in the text the original idea is lost because in the different sections EPC are not always mentioned.

I consider that the article is dispersed and sometimes confusing.  Some of my comments are:

It would be desirable to pay attention to the figures description, since it does not correlate with a text or image.  An example is Figure 4, where scheme indicates epigenetic reprograming and the text and figure description do not take it into account.

In the text when the authors refer different subpopulations and protocols to obtain EPC, it would be advisable to include the imunophenotipes  to show that they are different cells and not just nomenclatures.  

There are lines where the idea is not clear. For example 211 y 212 lines where it is not indicated if the cells arrive or leave the bone marrow.   In the line 221 it seems that all CD34+ cells are HSPC but it is important to mention if this cells come from bone marrow, peripheral blood or mobilized peripheral blood.  It is fundamental to note that not all CD34+ cells are stem or progenitor cells and its origin is always important

Reviewer 2 Report

This is an interesting, well written manuscript.

Comments/queries that the authors should consider.

1.      EPCs contribute to host defense response and the potential benefit of vascular precursor cell-based approaches for prevention and treatment of sepsis-induced vascular injury as well as vital organ system failure should be discussed.

2.      Exercise training has demonstrated beneficial effects on EPCs by increasing their number in peripheral circulation and improving their functional capacities in patients with heart failure. This should be commented.

3.      The potential effect of fine particulate matter (PM), which represents one of the main components of urban pollution, should be discussed.

4.      EPC-derived extracellular vesicles may exhibit therapeutic effects on several diseases, such as cardiovascular disease, acute kidney injury, acute lung injury, and sepsis. What do the authors believe.

5.      I believe that adding a subsection on the relationship between EPCs and cancer would greatly strengthen the manuscript.

6.      Some grammatical errors also in the figures.

Reviewer 3 Report

The review manuscript by Vinci et al. entitled "The Long Telling Story of "Endothelial Progenitor Cells": Where Are We at Now?" describes the journey of EPCs characterization, expansion, evaluation, and translation to clinical studies and administration. The manuscript is well-designed in a step-wise manner. The authors have discussed and considered all necessary aspects of EPCs characterization and potential therapeutic use. The article is supplemented with eye-catcher figures and informative tables. 

Round 2

Reviewer 1 Report

  The article has attended the previous suggestions

Reviewer 2 Report

No further comments